

# A 14 immune-related gene signature predicts clinical outcomes of kidney renal clear cell carcinoma

Yong Zou and Chuan Hu

Department of Oncology, The People's Hosipital of Hanchuan City, Hanchuan, Hubei, China

## ABSTRACT

Kidney renal clear cell carcinoma (KIRC) is the leading cause of kidney cancer-related deaths. Currently, there are no studies in tumor immunology investigating the use of signatures as a predictor of overall survival in KIRC patients. Our study attempts to establish an immune-related gene risk signature to predict clinical outcomes in KIRC. A total of 528 patients from The Cancer Genome Atlas (TCGA) database were included in our analysis and randomly divided into training ($n = 315$) and testing sets ($n = 213$). We collected 1,534 immune-related genes from the Immunology Database and Analysis Portal as candidates to construct our signature. LASSO-COX was used to find gene models with the highest predictive ability. We used survival and Cox analysis to test the model's independent prognostic ability. Univariate analysis identified 650 immune-related genes with prognostic abilities. After 1,000 iterations, we choose 14 of the most frequent and stable immune-related genes as our signature. We found that the signature was associated with M stage, T stage, and pathological staging. More importantly, the signature can independently predict clinical prognosis in KIRC patients. Gene Set Enrichment Analysis (GSEA) showed an association between our signature and critical metabolism pathways. Our research established a model based upon 14 immune-related genes that predicted the prognosis of KIRC patients based on tumor immune microenvironments.

## INTRODUCTION

Kidney cancer is one of the most common urological tumors worldwide, with approximately 403,262 new cases and 175,098 deaths associated with this form of cancer in 2018 (*Bray et al., 2018*). Kidney renal clear cell carcinoma (KIRC) is the leading cause of kidney cancer mortality and the most common type, accounting for 85% of kidney cancers (*Siegel, Miller & Jemal, 2017*; *Tseng, 2016*). Considering the aging population of KIRC patients and the rising expense of treatment, KIRC is gradually becoming the focus of geriatric cancer (*Tan, Filson & Litwin, 2015*). Despite the rapid development of cancer treatments, mortality rates in KIRC remain stagnant. With the progression of next generation sequencing and data mining techniques, it is urgent that we explore prognostic biomarkers for KIRC, using molecular characteristics and tumor immune environments to guide patient therapy.

Corresponding author
Chuan Hu, chuanhuv@163.com

Over the past decade our understanding of immune components, including the impact of tumor microenvironments on patient survival and therapy response (*Chen & Mellman, 2017*; *Grivennikov, Greten & Karin, 2010*), has increased. Some studies have found that tumor-infiltrating immune cells are able to serve as either tumor suppressors or promoters in microenvironments. For example, CD8$^+$ T cells have been associated with improved survival in cancer patients (*Gajewski, Schreiber & Fu, 2013*; *Governa et al., 2017*), while tumor associated macrophages and regulatory T cells demonstrate the ability to promote tumor development (*Nishikawa & Sakaguchi, 2014*; *Noy & Pollard, 2014*). Considering the complexity and significance of tumor immune microenvironments, it is imperative that we investigate immune-related biomarkers for KIRC patients. Recent studies have provided insight into the KIRC immune signature (*Geissler et al., 2015*; *Şenbabaoğlu et al., 2016*). *Şenbabaoğlu et al. (2016)* found mRNA signatures with the potential to be immunotherapeutic biomarkers in KIRC. However, their investigations do not include immune-related genes for analysis nor do they establish a systematic immune-related gene-risk signature for KIRC patients. *Khadirnaikar et al. (2019)* utilized immune associated lncRNA (long non-coding RNA) to construct prognostic subtypes in KIRC patients. Our immune clusters were more robust and independent. Additionally, they concentrated on lncRNA, not the overall immune-related genes. *Smith et al. (2019)* constructed endogenous retroviral signatures for KIRC patients, but they did not investigate the prognostic ability of the signature in different subtypes of patients. Therefore, it is essential that we explore a systematic prognostic signature based on tumor immune environments in KIRC.

In our study, we used RNA-seq data from The Cancer Genome Atlas (TCGA) to find immune-related genes with prognostic ability and to establish an immune-related risk signature for KIRC. To assess the clinical potential of the signature, we investigated the association between the signature, clinical parameters, and patient survival. Gene set enrichment analysis (GSEA) was performed to explore the molecular characteristics of the signature.

## MATERIALS & METHODS

### Patient cohort
The TCGA database from Xena browser was used to collect the clinical and RNA-seq data of 528 KIRC patients (*Goldman et al., 2019*). We randomly divided the dataset into training ($n = 315$) and test sets ($n = 213$). RNA-seq data was obtained to analyze the transcriptome profiling of RNA expression and were measured using fragments per kilobase of exon per million fragments mapped (FPKM). We performed a log2-based transformation to normalize RNA expression profiles. To ensure detection reliability, genes with more than half of their gene expression equal to zero were rejected in further investigations.

### Immune-related genes
We downloaded a comprehensive list of over 1,534 immune-related genes from the Immunology Database and Analysis Portal (ImmPort; https://immport.niaid.nih.gov; (*Bhattacharya et al., 2014*). These genes cover a diverse range of functions in immune-related pathways, including T cell receptor signaling pathways, B cell receptor signaling

pathways, antigen processing and presentation, chemokine, interleukins, interleukin receptors, chemokine receptors, cytokines, interferons, interferon receptors, cytokines receptors, natural killer cell cytotoxicity, tumor necrosis factor (TNF) family members, TNF family member receptors, transforming growth factor-b (TGF-b) family members, and TGF-b family member receptors.

## Establishment of immune-related gene signatures

We used the training set to establish an immune-related risk signature for KIRC and we performed univariate Cox analysis to screen out immune-related genes with prognostic properties (R package survival v2.42-6). After analyzing 1,534 immune-related genes, a total of 650 genes were identified with prognostic abilities. To identify the best gene model for predicting KIRC patient prognosis, genes with a *P*-value lower than 0.05 were evaluated using the Cox proportional hazards model with a lasso penalty (log lambda = −2.641, alpha = 1, iteration = 1,000, R package = "glmnet"; (*Friedman, Hastie & Tibshirani, 2010*). We performed a 10-fold cross-validation to estimate the penalty parameter in the training dataset. The gene model with the highest frequency in 1,000 iterations was chosen as the immune-related risk signature for KIRC. Linear weighing was performed on gene expression value and the Cox coefficient to determine the risk score (*Shang et al., 2017*).

## Assessment of prognostic ability

We used Harrell's c-index to estimate the predictive ability of the immune-related risk signature in the training, testing, and total cohort (*Harrell et al. 1982*). The independent prognostic ability of the immune-related risk signature was assessed using survival analysis as well as Cox analysis (R package survival, v2.42). We performed multivariable Cox regression to analyze the relationship between immune-related risk signatures and clinicopathological factors.

## Gene set enrichment analysis

To identify the biological functions and pathways between high- and low-risk groups, we conducted GSEA to investigate potential biological mechanisms in the Molecular Signatures Database (MSigDB; *Subramanian et al., 2005*). GSEA was performed using GSEApy, a Python wrapper for gene enrichment (https://pypi.org/project/gseapy/). We selected C2 and C5, including pathway databases and GO terms, from the MSigDB. The gesa sub-command of GSEApy was used in GSEA with default parameters. Enriched gene sets with a false discovery rate (FDR) of less than 0.25 and a *P*-value of less than 0.05 were considered statistically significant.

## Statistical analysis

Boxplots were created with the R package, ggplot2 (v3.0.0). The R package ComplexHeatmap (v1.18.1) was used to create heatmaps (*Gu, Eils & Schlesner, 2016*). We counted C-index with R packages "survcomp" (*Harrell Jr et al., 1982*; *Schröder et al., 2011*). The student's *t*-test was performed for statistical comparison. We chose R to conduct statistical analysis (https://www.r-project.org/). *P*-values lower than 0.05 were considered statistically significant. The main code of analysis was pushed to github (https://github.com/huchua/KIRC).

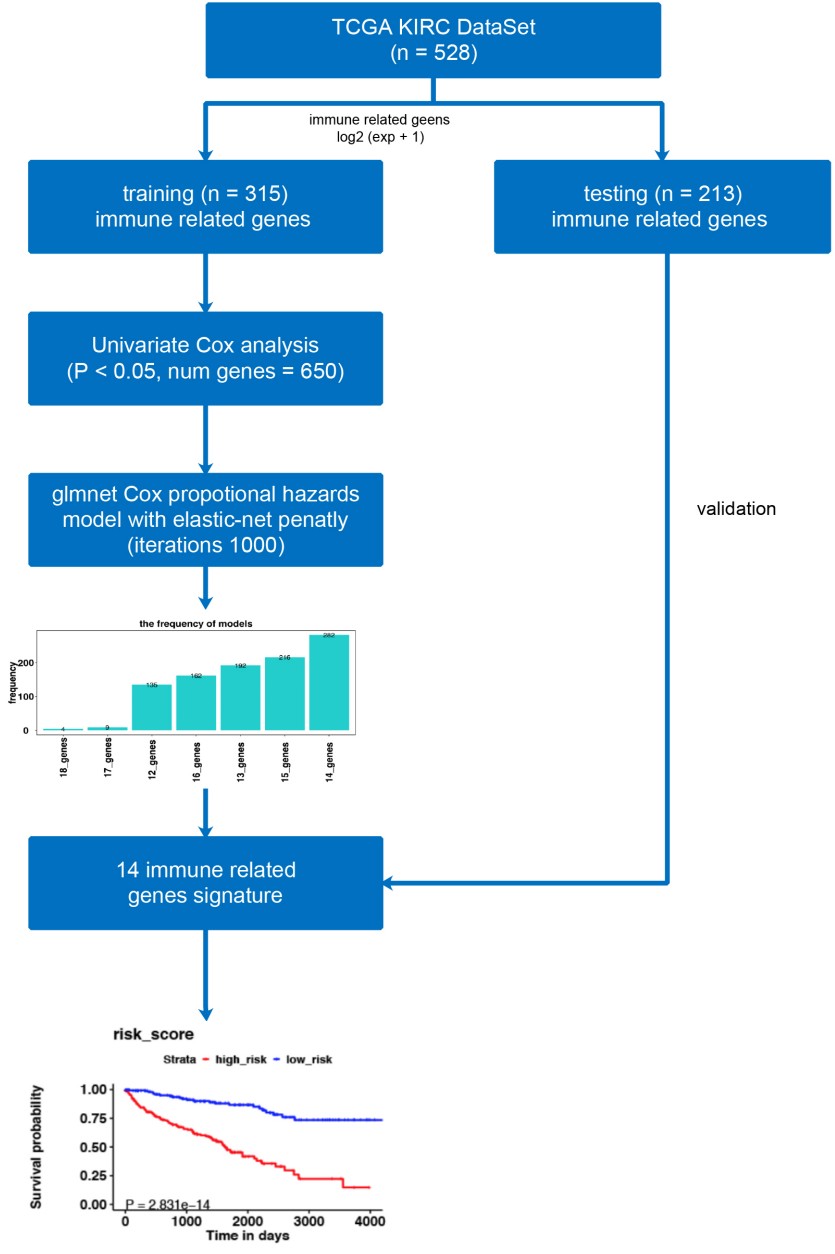

**Figure 1** **The workflow of model construction.** Our workflow constructing the model for risk-score signatures.

## RESULTS

### Establishment and validation of the immune-related gene signature in KIRC

Figure 1 illustrates the workflow we used to develop an immune-related gene-risk signature. The immune-related gene signature was constructed within the KIRC training data set, while we applied the testing set to validate the signature. After 1,000 iterations, seven unique

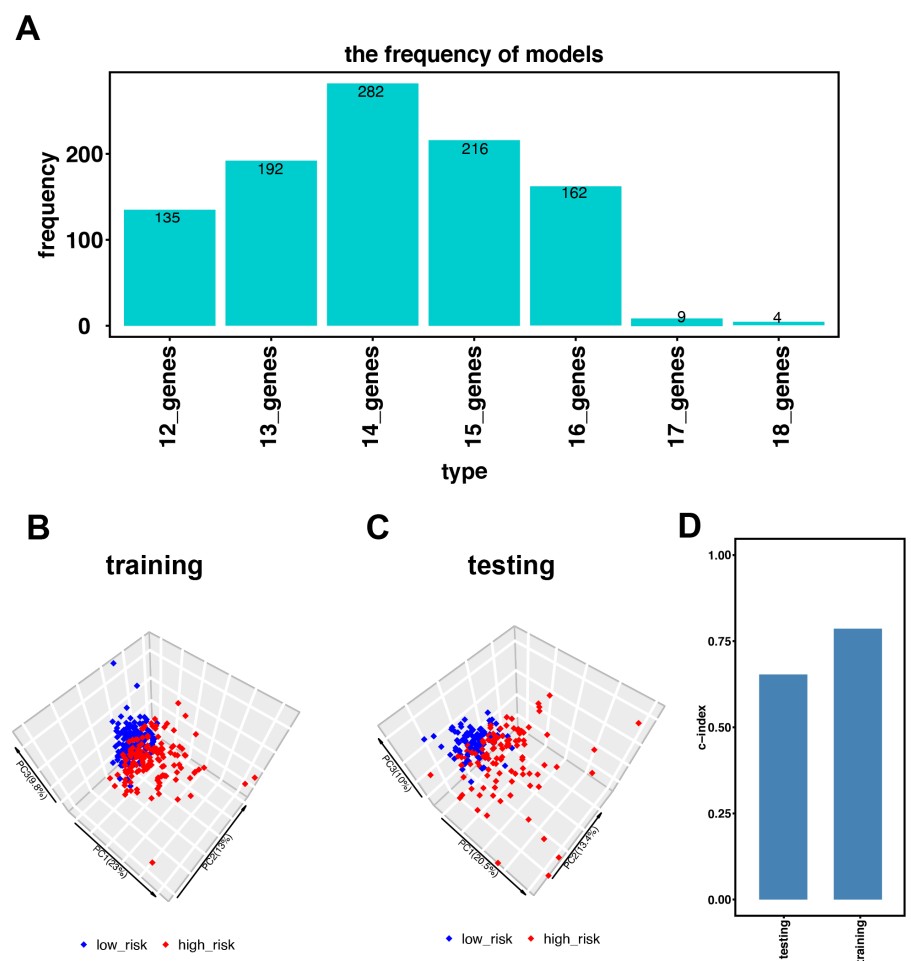

**Figure 2** **Establishment of the immune-related gene-risk signature for KIRC.** (A) After 1,000 iterations, seven gene model candidates for constructing the signature. The 14 immune-related gene-risk signature ranked highest among all gene models. (B–C) Principle components analysis (PCA) using our risk signature in training and testing group. Red dots represent high-risk patients while blue dots represent low-risk patients. (D) In training and testing cohort, the c-index is 0.7862 and 0.6534, respectively. ($P < 0.001$).

gene models were selected (Fig. 2A, Table S1). The model selected as the immune-related gene-risk signature consisted of 14 genes and ranked the highest in frequency 282 times. Parameter selection in the LASSO-cox model was log lambda $= -2.641$ and alpha $= 1$. The univariate and multivariate Cox analysis of the 14 immune-related genes are shown in Tables 1 and 2. The principle component analysis of the 14 immune-related gene signature displayed a different distribution pattern between low- and high-risk groups when comparing the training and testing (Figs. 2B–2C). This indicated that the low- and high-risk groups had different immune phenotypes. In the training and testing, the c-index was 0.7862 and 0.6534, respectively ($P < 0.001$; Fig. 2D). In a time-dependent receiver operating characteristic curve (ROC) created for training and testing datasets, area under the curve (AUC) values at 1, 3, and 5 years were 0.679, 0.63, 0.627 and 0.65, 0.596, 0.568, respectively (Fig. S2). The 14 immune-related genes are AR, BID, BMP8A, CCL7,

**Table 1  Univariate Cox analysis of 14 immune relate genes in all cohort.**

| Variable | HR | 95% CI | *p* value |
|---|---|---|---|
| KLRC2 | 20.657 | 9.116–46.81 | <0.001 |
| CCL7 | 2.757 | 2.090–3.637 | <0.001 |
| SEMA3A | 1.748 | 1.437–2.126 | <0.001 |
| SEMA3G | 0.617 | 0.534–0.714 | <0.001 |
| CCR10 | 1.759 | 1.381–2.241 | <0.001 |
| BMP8A | 1.813 | 1.298–2.532 | <0.001 |
| FGF17 | 7.335 | 3.202–16.803 | <0.001 |
| GDF1 | 1.483 | 1.062–2.070 | 0.021 |
| IL4 | 109.106 | 34.328–346.781 | <0.001 |
| LHB | 5.053 | 3.240–7.881 | <0.001 |
| TXLNA | 2.329 | 1.500–3.618 | <0.001 |
| AR | 0.548 | 0.463–0.649 | <0.001 |
| IL20RB | 1.225 | 1.145–1.310 | <0.001 |
| BID | 3.505 | 2.439–5.035 | <0.001 |

CCR10, FGF17, GDF1, IL20RB, IL4, KLRC2, LHB, SEMA3A, SEMA3G, and TXLNA. K-M (Kaplan–Meier) survival curves and gene expression of the 14 immune-related genes are shown in Fig. 3 and Fig. S3. Because of the differences between the log-rank test and the univariate cox analysis, the results of the univariate cox analysis of other genes except GDF1 are very consistent with the results of the K-M (Kaplan–Meier) survival curves.

## Correlation between the signature of immune-related genes and clinical parameters

Among the 14 immune-related genes, four genes (TXLNA, SEMA3G, AR, and BID) had a high expression, and 10 genes (IL20RB, CCR10, BMP8A, SEMA3A, CCL7, GDF1, KLRC2, LHB, FGF17, and IL4) had a low expression (Figs. 4A, 4B). The relationship between the signature and clinical factors demonstrated that patients with advanced pathological staging, M stage, and T stage had a higher risk score than those with early stage disease (Figs. 4A, 4B; 5A, 5B, 5D). However, we did not find a correlation between the signature and N stage (Figs. 4A, 4B; 5C).

## Influence of the immune-related gene signature on patient prognosis

We then assessed whether this signature influenced KIRC patient prognosis. Survival analysis showed that patients with a high-risk score were associated with poor survival outcomes in the training and testing (Figs. 4C; 4E). We found that the signature also predicted survival outcomes in subgroups of KIRC patients, including stage I-II (Fig. 6A), stage III-IV (Fig. 6B), M0 stage (Fig. 6C), M1 stage (Fig. 6D), N0 stage (Fig. 6E), N1 (Fig. 6F), T1 (Fig. 6G), T2 (Fig. 6H), T3 (Fig. 6I), and T4 (Fig. 6J). Multivariate Cox analysis revealed that the risk signature was able to independently predict overall survival in KIRC patients (Figs. 4D, 4F, Table 3).

| Table 2 | Multivariate Cox analysis of 14 immune relate genes in all cohort. | | | | | | |
|---|---|---|---|---|---|---|---|
| **AR** | 0.638 | 0.556–0.731 | <0.001 | **IL20RB** | 1.268 | 1.102–1.46 | 0.001 |
| age | 1.352 | 1.143–1.599 | <0.001 | age | 1.414 | 1.195–1.672 | <0.001 |
| stage | 3.725 | 2.708–5.124 | <0.001 | stage | 3.275 | 2.351–4.562 | <0.001 |
| gender | 0.943 | 0.687–1.294 | 0.718 | gender | 1.025 | 0.744–1.412 | 0.879 |
| **BID** | 1.411 | 1.211–1.645 | <0.001 | **IL4** | 1.458 | 1.294–1.643 | <0.001 |
| age | 1.407 | 1.188–1.666 | <0.001 | age | 1.39 | 1.171–1.651 | <0.001 |
| stage | 3.173 | 2.284–4.41 | <0.001 | stage | 3.426 | 2.484–4.725 | <0.001 |
| gender | 1.039 | 0.755–1.43 | 0.814 | gender | 0.999 | 0.726–1.375 | 0.997 |
| **BMP8A** | 1.213 | 1.057–1.391 | 0.006 | **KLRC2** | 1.402 | 1.233–1.594 | <0.001 |
| age | 1.454 | 1.229–1.721 | <0.001 | age | 1.383 | 1.169–1.637 | <0.001 |
| stage | 3.607 | 2.619–4.969 | <0.001 | stage | 3.429 | 2.483–4.737 | <0.001 |
| gender | 1.052 | 0.765–1.447 | 0.754 | gender | 0.958 | 0.694–1.322 | 0.792 |
| **CCL7** | 1.329 | 1.202–1.471 | <0.001 | **LHB** | 1.285 | 1.168–1.414 | <0.001 |
| age | 1.455 | 1.226–1.728 | <0.001 | age | 1.393 | 1.178–1.647 | <0.001 |
| stage | 3.485 | 2.529–4.802 | <0.001 | stage | 3.519 | 2.55–4.854 | <0.001 |
| gender | 1 | 0.726–1.377 | 1 | gender | 1.17 | 0.844–1.622 | 0.347 |
| **CCR10** | 1.235 | 1.111–1.372 | <0.001 | **SEMA3A** | 1.214 | 1.107–1.33 | <0.001 |
| age | 1.382 | 1.166–1.639 | <0.001 | age | 1.445 | 1.218–1.715 | <0.001 |
| stage | 3.844 | 2.793–5.292 | <0.001 | stage | 3.549 | 2.572–4.898 | <0.001 |
| gender | 1.034 | 0.751–1.424 | 0.836 | gender | 1.108 | 0.802–1.529 | 0.535 |
| **FGF17** | 1.235 | 1.112–1.372 | <0.001 | **SEMA3G** | 0.652 | 0.549–0.775 | <0.001 |
| age | 1.399 | 1.181–1.657 | <0.001 | age | 1.379 | 1.164–1.633 | <0.001 |
| stage | 3.722 | 2.705–5.121 | <0.001 | stage | 3.29 | 2.384–4.539 | <0.001 |
| gender | 1.109 | 0.806–1.525 | 0.527 | gender | 0.973 | 0.707–1.339 | 0.867 |
| **GDF1** | 1.086 | 0.975–1.211 | 0.135 | **TXLNA** | 1.285 | 1.098–1.505 | 0.002 |
| age | 1.411 | 1.191–1.672 | <0.001 | age | 1.433 | 1.208–1.7 | <0.001 |
| stage | 3.76 | 2.735–5.17 | <0.001 | stage | 3.696 | 2.687–5.085 | <0.001 |
| gender | 1.103 | 0.796–1.529 | 0.555 | gender | 1.212 | 0.87–1.688 | 0.255 |

## Engagement of the immune-related gene risk signature in biological pathways and functions

GSEA was used to investigate the signature's biological pathways and functions. There were 177 KEGG pathways and 4,528 GO terms used in our investigation. Our analysis found that the signature was able to engage in a total of 19 enriched KEGG pathways (FDR < 0.25; Table 4). The low-risk signature was significantly correlated with 10 pathways, including the citrate cycle (TCA cycle) pathway, fatty acid metabolism pathway, propanoate metabolism pathway, butanoate metabolism pathway, peroxisome pathway, lysine degradation pathway, valine leucine and isoleucine degradation pathway, proximal tubule bicarbonate reclamation pathway, vasopressin regulated water reabsorption pathway, and the pyruvate metabolism pathway (Table 4). Similarly, eight GO annotations were enriched in the low-risk group (FDR < 0.25; Table 4).

## DISCUSSION

Our study used the TCGA database and immune-related genes to establish a KIRC signature consisting of 14 immune-related genes. We found that patients in the

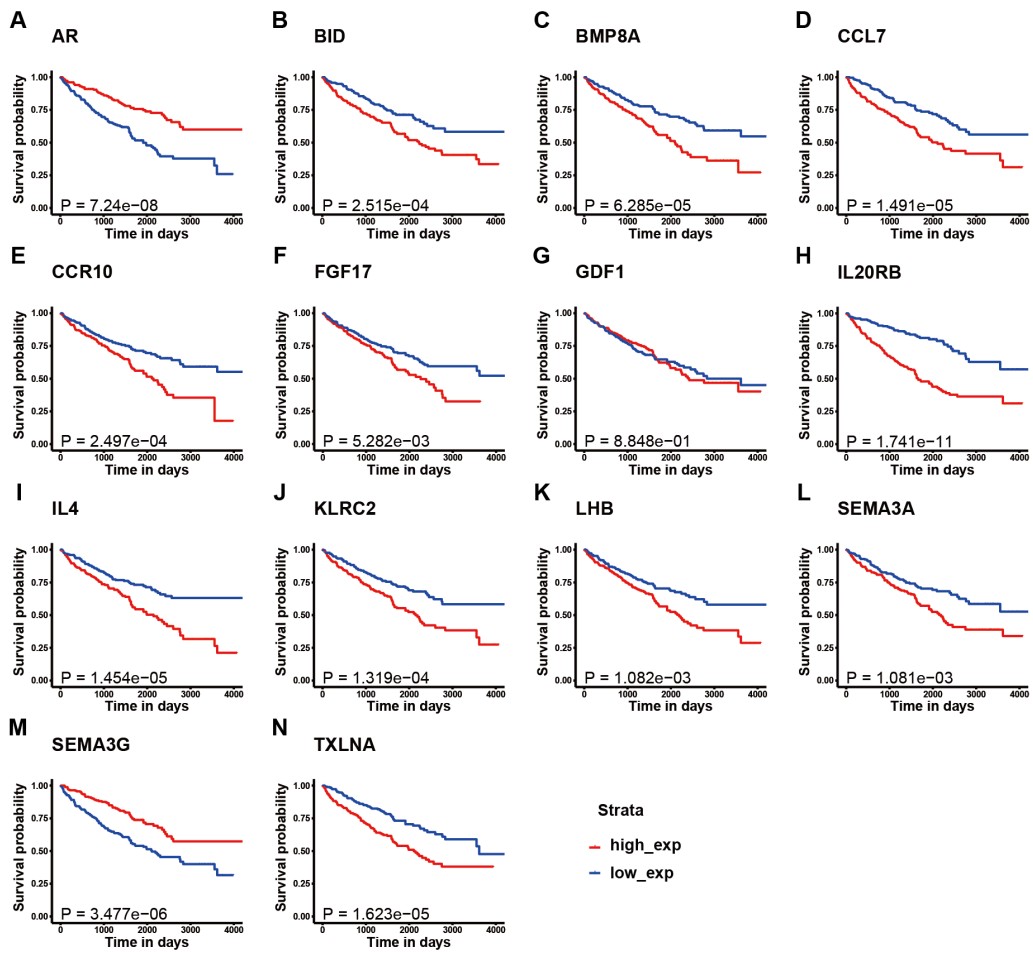

**Figure 3  K–M survival curve analysis of the 14 immune-related genes.** K–M survival curve analysis of the 14 immune-related genes used to establish the signature for KIRC, including (A) AR, (B) BID, (C) BMP8A, (D) CCL7, (E) CCR10, (F) FGF17, (G) GDF1, (H) IL20RB, (I) IL4, (J) KLRC2, (K) LHB, (L) SEMA3A, (M) SEMA3G, and (N) TXLNA. (A, M) The patients with high expression of AR and SEMA3G had better prognosis; (B-F, H-L, N) The patients with high expression of these gene had poor survival; (G) There is no obvious correlation between the expression of GDF1 and the prognosis of patients.

high-risk group showed a positive association with M stage, T stage, and advanced pathological staging. Additionally, the signature exhibited strong prognostic abilities and independently predicted KIRC patient prognosis. Functional analysis highlighted the significance of our signature based on its involvement in many important pathways.

In our investigation, we used a total of 14 immune-related genes to construct our signature. In the signature, CCL7 increased the peripheral blood mononuclear cell recruitment in renal cell cancer through the inhibition of let-7d (*Riihimaki et al., 2014*). *Wyler et al. (2014)* demonstrated the ability of CCL7 to recruit monocytes through CCR2, promoting renal cell cancer metastasis to the brain. This indicates that CCL7 is a major factor in the development of KIRC in tumor immune microenvironments

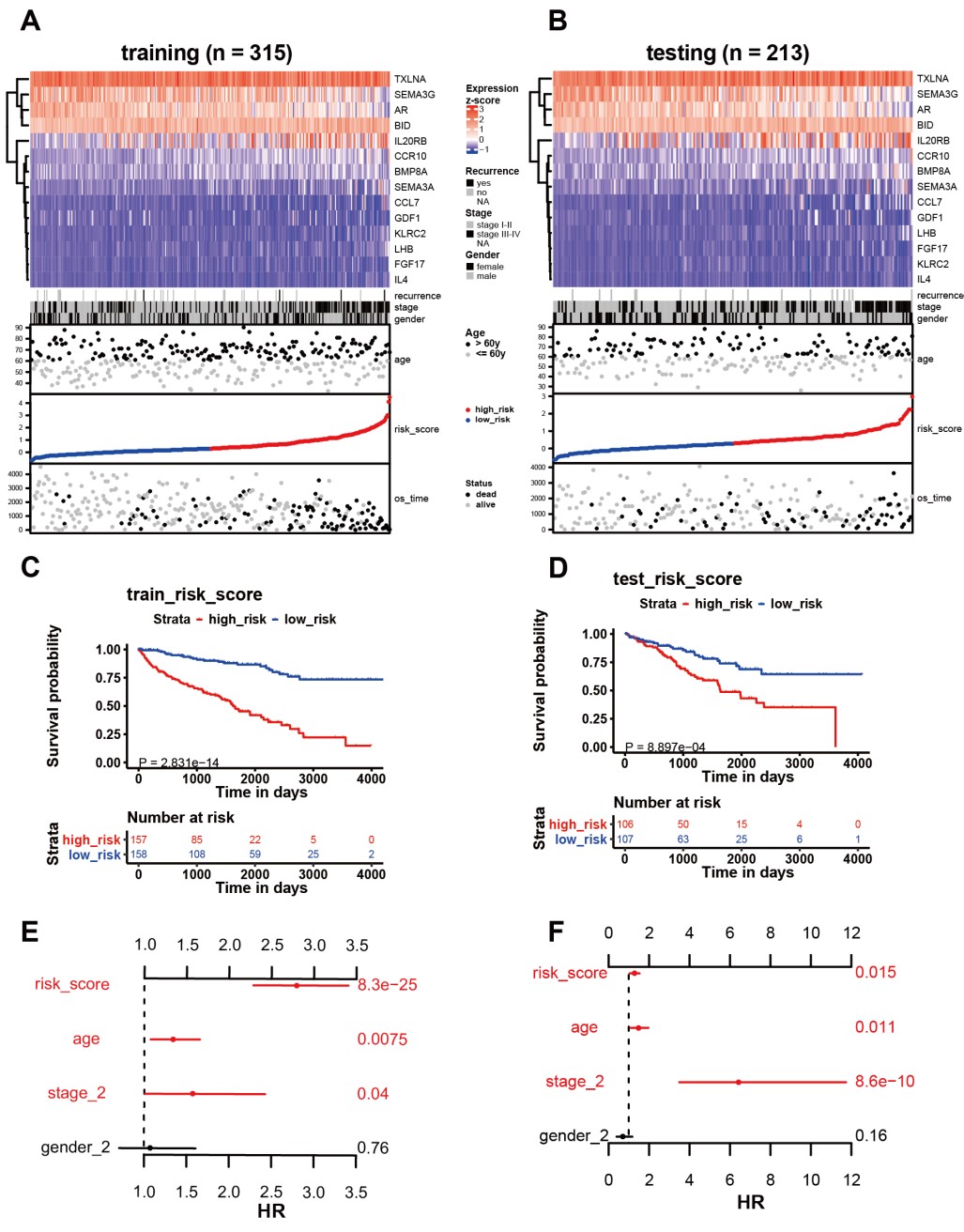

**Figure 4 Analysis of the 14 immune-related genes and their predictive ability in both training and testing cohorts.** (A, B) As demonstrated by the heatmap, the expression distribution of the 14 immune-related genes differed from each other in training and testing cohorts. Each column represents the same patient and corresponds to the point below showing risk score distribution, survival status, and time in KIRC patients. Each point represented one patient sorted by the rank of the risk score. Patients with high-risk, low-risk, deceased, and alive were marked by red, blue, black, and grey, respectively. The advanced-stage patients showed high-risk scores in training and testing cohorts without stratification. (C, E) Survival analysis showed that patients with high-risk scores correlated with poor survival outcomes. (D, F) The multivariate Cox analysis in training and testing cohorts. The 14 immune-related gene signature was able to serve as an independent prognostic factor for OS.

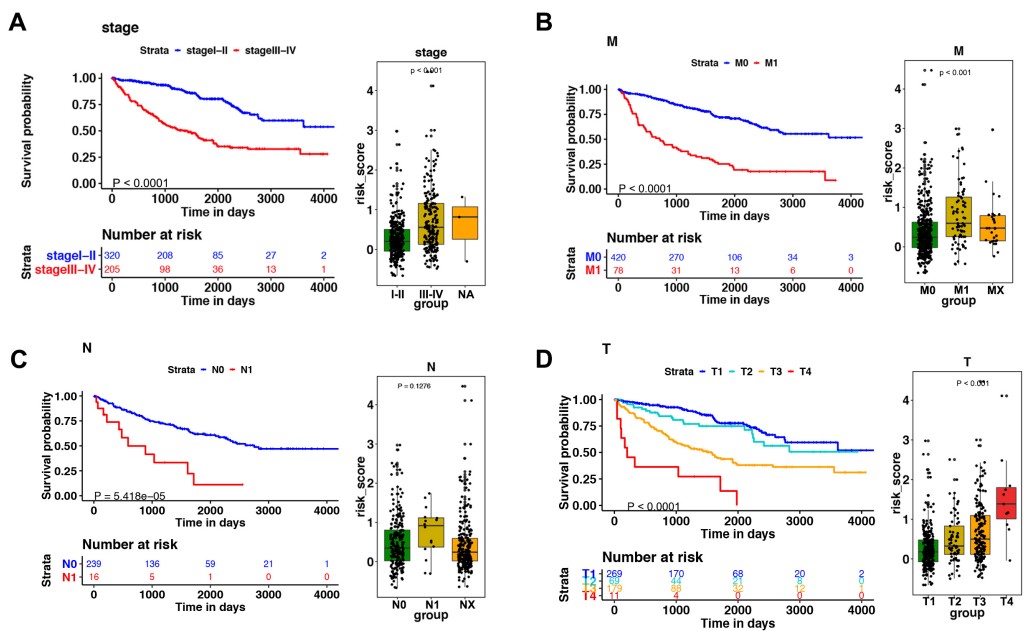

**Figure 5  The relationship between the risk score and clinical factors.** The K–M survival curve of (A) pathological staging, (B) M stage, (C) N stage, and (D) T stage. The boxplots demonstrated the relationship between the risk score and (A) pathological staging, (B) M stage, (C) N stage, and (D) T stage. Those with high levels of T, M stage, and advanced pathological staging presented a higher risk score.

and may be a potential immunotherapy target. Fibroblast growth factor receptor 17 (FGF17) is another immune-related gene in our signature. FGF17 demonstrates a variety of functions in cancer development. *Gauglhofer et al. (2011)* found that FGF17 was involved in the paracrine and autocrine signaling of hepatocellular carcinoma and promoted the neoangiogenesis of hepatocellular carcinoma. Heer et al. (2004) showed that FGF17 was overexpressed in prostate cancer and participated in prostate carcinogenesis. Our study showed that FGF17 plays an important role in KIRC based on tumor immunology. However, the underlying mechanism of FGF17 in KIRC immune microenvironments requires further investigation. In our signature, IL-4 is another well-studied immune-related biomarker for KIRC. IL-4, released by immune cells, controls the expression of B7-H1, thus altering T cell responses in KIRC (*Quandt et al., 2014*). The investigation conducted by Chang et al. (2015) demonstrated that IL-4 expression can predict KIRC patient recurrence and survival outcomes. Collectively, these investigations reinforce the significance of the 14 immune-related gene-risk signature in KIRC.

Our signature is associated with the survival outcome of KIRC patients and clinical parameters, including pathological staging, M stage, and T stage. No correlation was found between the signature and N-stage, possibly due to a lack of N-stage information for many patients. According to the immune-related gene-risk signature, our study found that clinical cohorts in KIRC have different immune-related risk factors. This signature also reflects differences in tumor immune

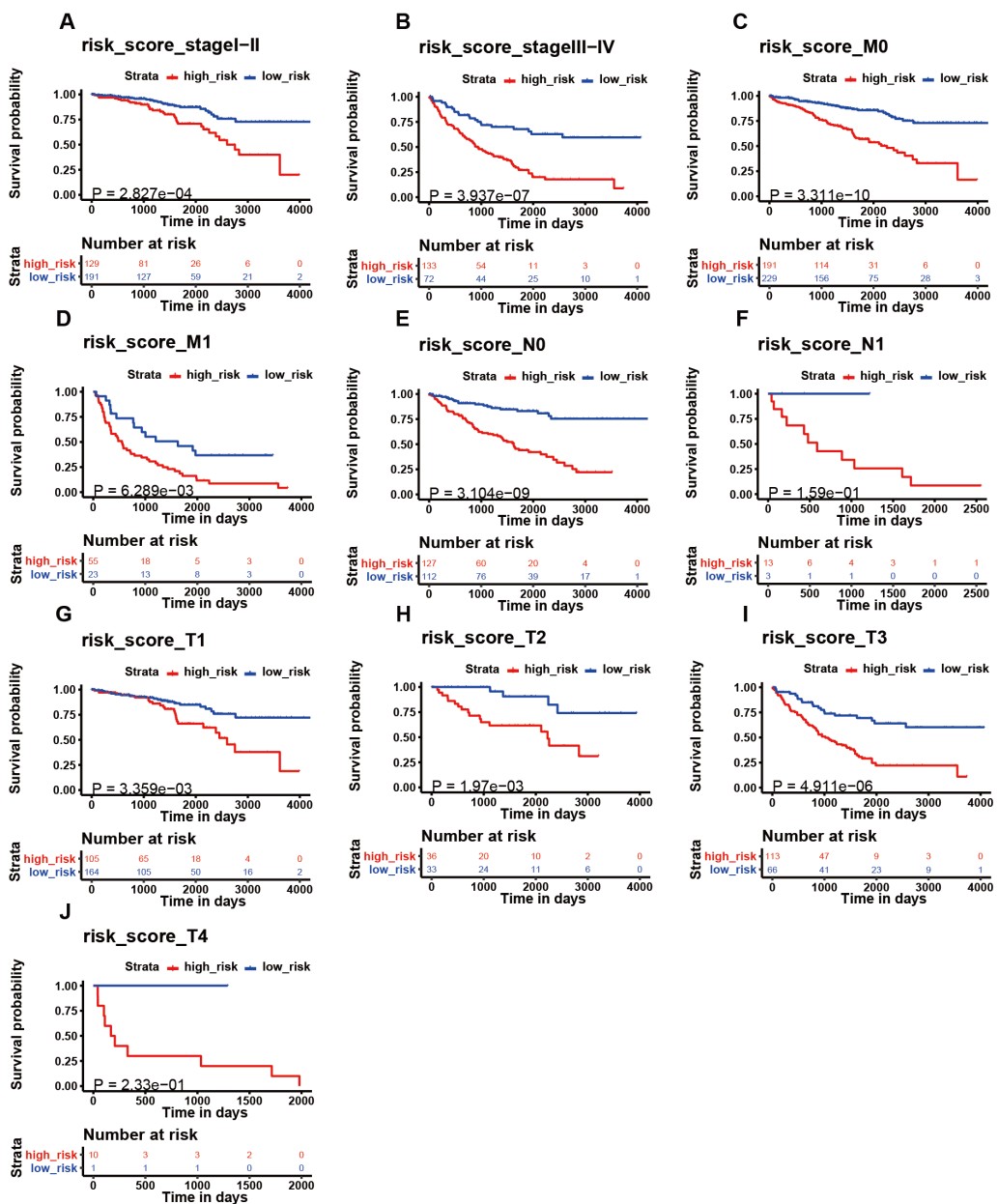

**Figure 6** **The K–M analysis of the risk signature in subgroups of patients.** The K–M analysis of the risk signature in subgroups of patients with stage I–II (A), stage III–IV (B), M0 stage (C), M1 stage (D), N0 stage (E), N1 stage (F), T1 stage (J), T2 stage (H), T3 stage (I), and T4 stage (J).

microenvironments and predicts survival outcomes in KIRC patients; thus, demonstrating the clinical significance of our signature and its possible use as a survival predictor in KIRC.

Despite of the relatively low C-index of our testing set (0.6534), the C-index of the testing set falls within a similar range as those of other related studies. For instance, in the study of Bailiang Li et al., they developed an immune signature in non-small

**Table 3  Univariate and Multivariate Cox analysis of risk and clinical parameters in training and testing.**

|          | Variable   | Univariate | | | Multivariate | | |
|----------|------------|-------|--------------|---------|-------|--------------|---------|
|          |            | HR    | 95% CI       | *p* value | HR    | 95% CI       | *p* value |
| training | risk_score | 4.484 | 3.508–5.732  | <0.001  | 2.789 | 2.294–3.391  | <0.001  |
|          | age        | 1.028 | 1.011–1.045  | 0.001   | 1.334 | 1.080–1.648  | 0.007   |
|          | stage      | 3.010 | 2.030–4.462  | <0.001  | 1.567 | 1.021–2.405  | 0.040   |
|          | gender     | 0.984 | 0.663–1.460  | 0.937   | 1.066 | 0.713–1.594  | 0.756   |
| testing  | risk_score | 2.010 | 1.490–2.711  | <0.001  | 1.262 | 1.046–1.522  | 0.015   |
|          | age        | 1.027 | 1.007–1.047  | 0.007   | 1.460 | 1.091–1.954  | 0.011   |
|          | stage      | 5.954 | 3.396–10.439 | <0.001  | 6.406 | 3.539–11.598 | <0.001  |
|          | gender     | 0.867 | 0.520–1.445  | 0.584   | 0.682 | 0.401–1.161  | 0.159   |

cell lung cancer with the C-index of 0.64 (*Li et al., 2017*). Besides, our signature achieved a similar C-index of testing set with the immune signature in ovarian cancer (0.625) (*Shen et al., 2019*). More importantly, the C-index of our testing set showed a similar accuracy with the C-index of clinical staging systems in renal cancer (0.62) (*Qu et al., 2018*).

Our study expands on the signatures association with several important pathways, especially the metabolism pathway. This may reflect the mutual interaction between tumor metabolism and tumor immunology in KIRC. *Pearce et al. (2013)* showed that metabolic reprogramming can influence the fate and function of T cells in tumors. Our study further indicates the importance of metabolism pathways in KIRC immune microenvironments.

We acknowledge the limitation of our evaluation scheme, including randomly dividing the full data set into training and testing data set, which result in the inherent bias for the specific study. Another limitation of our research is the lack of independent validation data sets in our evaluation scheme. Therefore, our study should be further validated through a prospective cohort data to further illustrate the robustness of the model.

## CONCLUSIONS

This investigation utilized RNA-seq data from the TCGA database to construct a 14 immune-related gene-risk signature with the ability to independently predict survival outcomes in KIRC; thus, providing novel clinical applications and possible immune targets for KIRC.

**Table 4  GSEA analysis.** Gene set enrichment analysis for high- and low-risk groups, using 177 KEGG pathways and 4,528 GO terms.

| Type | Term | NES | *P*-value | FDR |
|------|------|-----|-----------|-----|
| KEGG | Citrate cycle tca cycle | −1.957 | <0.001 | 0.076 |
| | Fatty acid metabolism | −1.863 | <0.001 | 0.079 |
| | Propanoate metabolism | −1.883 | <0.001 | 0.084 |
| | Butanoate metabolism | −1.866 | 0.022 | 0.089 |
| | Peroxisome | −1.913 | <0.001 | 0.093 |
| | Lysine degradation | −1.891 | 0.011 | 0.096 |
| | Valine leucine and isoleucine degradation | −2.001 | <0.001 | 0.110 |
| | Proximal tubule bicarbonate reclamation | −1.958 | <0.001 | 0.114 |
| | Vasopressin regulated water reabsorption | −1.789 | 0.030 | 0.144 |
| | Pyruvate metabolism | −1.802 | 0.012 | 0.146 |
| | Prostate cancer | −1.763 | <0.001 | 0.149 |
| | Terpenoid backbone biosynthesis | −1.768 | 0.022 | 0.156 |
| | Endometrial cancer | −1.729 | 0.011 | 0.162 |
| | Glycolysis gluconeogenesis | −1.735 | 0.033 | 0.169 |
| | Glycosylphosphatidylinositol gpi anchor biosynthesis | −1.701 | 0.031 | 0.181 |
| | Biosynthesis of unsaturated fatty acids | −1.679 | 0.048 | 0.202 |
| | Epithelial cell signaling in helicobacter pylori infection | −1.671 | 0.012 | 0.203 |
| | Tryptophan metabolism | −1.654 | 0.032 | 0.215 |
| | Adherens junction | −1.622 | 0.052 | 0.250 |
| GO | Branched chain amino acid metabolic process | −2.058 | <0.001 | 0.160 |
| | Tricarboxylic acid metabolic process | −2.036 | <0.001 | 0.172 |
| | Myelin sheath | −2.061 | <0.001 | 0.178 |
| | Positive regulation of viral release from host cell | −2.040 | <0.001 | 0.183 |
| | Oxidoreductase activity acting on the aldehyde or oxo group of donors nad or nadp as acceptor | −2.068 | 0.001 | 0.191 |
| | Lipid oxidation | −2.012 | 0.003 | 0.219 |
| | Microbody membrane | −2.070 | 0.001 | 0.233 |
| | Microbody lumen | −1.997 | 0.003 | 0.242 |

### Funding

The authors received no funding for this work.

### Competing Interests

The authors declare there are no competing interests.

### Author Contributions

- Yong Zou conceived and designed the experiments, performed the experiments, analyzed the data, prepared figures and/or tables, authored or reviewed drafts of the paper, and approved the final draft.
- Chuan Hu performed the experiments, analyzed the data, authored or reviewed drafts of the paper, and approved the final draft.

## Data Availability

The data are available at UCSC Data Hubs (https://xenabrowser.net/hub/). The link of each data file is as follows: 1. https://gdc.xenahubs.net/download/TCGA-KIRC.htseq_fpkm.tsv.gz; 2.https://gdc.xenahubs.net/download/TCGA-KIRC.GDC_phenotype.tsv.gz ; 3.https://gdc.xenahubs.net/download/TCGA-KIRC.survival.tsv.gz

The main code of the analysis is available at GitHub: https://github.com/huchua/KIRC.

## Supplemental Information

Supplemental information for this article can be found online at http://dx.doi.org/10.7717/peerj.10183#supplemental-information.

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
