# Peer review of "A 14 immune-related gene signature predicts clinical outcomes of kidney renal clear cell carcinoma"

_PeerJ, doi:10.7717/peerj.10183_

## Round 0.1 · original submission · Major Revisions

Three reviewers have examined your manuscript. I agree with them that significant improvements need to be made to this manuscript.

Reviewer 1 mentions lack of novelty. PeerJ does not require novelty, but it does require that you do more to place the article in context with prior studies that have been performed. "The article should include sufficient introduction and background to demonstrate how the work fits into the broader field of knowledge. Relevant prior literature should be appropriately referenced." The Introduction does touch on this briefly, but a more careful and thorough evaluation of prior work should be provided. What specifically does your article provide that others did not? Or is it a replication study?

Reviewer 1 mentions that the authors did not provide any of their own data. This is okay. However, as all the reviewers mentioned, when proposing a gene signature for potential clinical relevance, there is a need to validate the signature on independent datasets. Reviewers 2 and 3 mentioned that you might be able to find these in Gene Expression Omnibus. If you do this, make sure to keep those datasets completely independent from the TCGA data.

Two reviewers mentioned that the English language usage needs work. I agree with them. PeerJ's criteria state, "The article must be written in English and must use clear, unambiguous, technically correct text. The article must conform to professional standards of courtesy and expression." I know this is difficult because English is not your first language, but it must be improved in many places throughout the manuscript.

Please address the other comments from Reviewers 2 and 3.

In addition, below are a few of my own comments.

* The Methods section needs more detail. For example, it says, "univariate analysis was performed" but it is hard to know exactly what that means. Later it indicates "lambda = lambda.min" but it doesn't indicate what lambda.min equals. Few details are provided on specifically how the GSEA analysis was performed.

* I'm not sure that I understand exactly how model selection was performed. For example, this language is unclear to me: "A total of 650 immune-related genes went through the cox proportional hazards regression with 10-fold cross112 validation so as to establish an immune-related gene risk signature for KIRC. After 1,000 iterations, a total of seven 113 models consisting of different number of genes were arranged."

* The "Establishment and validation of the signature of immune-related gene in KIRC" section is confusing because it comes after other details have been provided about methods. These sections should be integrated together.

* It's commendable that the authors provided a GitHub repository with analysis code and specifically mention versions of R packages used. However, to make it easier for others to verify your code, please do the following:
- Provide instructions on installing packages or else use something like MyBinder.org.
- Provide data files together with the code (or else a specific URL for each data file).

* Cite Harrell's c-index

* In many of the figures, the text is very small, making it difficult to read. In many figures, axis labels use underscores (should be replaced with space characters) or acronyms that are undefined.

* Did you use clinical covariates in the Cox proportional hazards models? If not, why?

There are likely additional changes that will need to be made. But I hope these will be helpful as you revise.

Reviewer 1 ·

Basic reporting

/

Experimental design

/

Validity of the findings

/

Additional comments

A 14-immune-related-gene signature predicts clinical outcome of kidney renal clear cell carcinoma - for the journal PeerJ (#43459)

Comments:
Thank you for the paper. This article suffers from several major flaws:
1. The lack of novelty
The clinic value of immune-related genes in a wide variety of human malignancies, including kidney renal clear cell carcinoma, had been extensively investigated and clearly elucidated in a great number of previous studies. The authors have failed to list the specific novel approaches and/or findings from their study compared to those of previous studies.

2. The lack of any original datum from authors’ own studies
All the experimental data presented and analyzed in the current study were downloaded from a publicly available database. The authors of the current study have failed to contribute any original laboratory or clinic finding.

3. The lack of evidence
This is a simple replication of findings previously published in another tumor model. The conclusions are based on a single cohort. More independent cohorts needed to be used for the validation.

Reviewer 2 ·

Basic reporting

the language is poor and needs English editing.

Experimental design

No comments

Validity of the findings

No comments.

Additional comments

The study adopted the TCGA data and performed the bioinformatics analysis. First, the aim of the study is clear, the methods are clearly explained, and the results are interesting. But, the Discussion section is not adequate to discuss the results, and needed to do in-depth analysis.
Major revision:
1. The main limitation of this study is that the language is poor and needs English editing, because the manuscript had many grammar mistakes.
2. The author had better to add some datasets to verify the result robustness, such as GEO dataset.

Reviewer 3 ·

Basic reporting

no comment

Experimental design

In this manuscript, Dr Yong Zou et.al established a 14-immune-related gene model predicting prognosis of KIRC patients based on tumor immune microenvironment. They also investigated KEGG pathways and GO terms correlated with the signature. In summary, the article may provide novel clinical immune targets. While, there are still some concerns as follows:

1) The authors should analyze expression of the 14 immune genes in tumor and normal tissue.
2) The authors should present a time-dependent ROC curve .(1 year,3year and 5 year)
3) The authors should use extra data to verify the accuracy of the model (GEO datasets)

Validity of the findings

no comment

---

## Round 0.2 · Major Revisions

I apologize for taking awhile. Two of the original reviewers were unavailable to re-review the article. So I sent it out for review to a new reviewer. As you can see below, this reviewer had some concerns. Please address this reviewer's concerns about the C-index values. At a minimum, add some discussion to the paper about the potential clinical relevance (or lack thereof) of a C-index in this range.

The reviewer expressed concern about the survival curves crossing in Figure 3 for the GDF1 gene. Please address that briefly. But more importantly, these p-values should be adjusted for multiple tests.

Regarding the GSEA analysis, an FDR threshold of 0.25 is fairly high. A threshold in the range of 0.05 to 0.20 is more customary. Please modify this or provide a short justification for using 0.25.

Per the reviewer's comment, please review the references and update them to more recently published articles in the field.

Some of the text is still much too small in Figures 3, 4, 6, 7, 8, and 9.

I'm not sure that it makes sense to include the figures from the GSEA output. Consider providing a table with the GSEA results rather than showing these graphics in Figures 8 and 9.

Thank you for providing slightly more detail in the GitHub repository about which packages to install. However, instead of explaining this verbally, give the reader the exact code they need to install the packages. Also, your R code should download the data files. The idea is that someone could re-run your analysis without having to modify your code.

Reviewer 3 ·

Basic reporting

No more concern.

Experimental design

No more concern.

Validity of the findings

No more concern.

Additional comments

The authors have addressed all my concerns.

Reviewer 4 ·

Basic reporting

Dr Zou et al. reported that they identified a combination of 14 immune-related genes that could predict clinical outcomes of kidney renal cell carcinoma in the TCGA data set.
This study was interesting, but there were several points to be mentioned.

Experimental design

In this study, mRNA expression data from only the TCGA cohort was mainly used though the author divided them into 2 groups as training and testing sets. The reviewer recommends that they would variate the usefulness of 14 immune-related genes using the other data set.

Validity of the findings

In Fig.2, the authors showed the data of PCA analysis of each data set. However, the sum of the explanatory power of the three principle components was less than 50% within all the groups. Also, the C-index of the testing sets was 0.6534, and it was statistically not useful as a predictive marker. The authors should clarify the reason they considered the combination of 14 immune- related genes as a useful predictive marker.

In Fig.3, the survival curve of the higher GDF1 group crossed that of the lower GDF1 group. Although there was a significant difference, the authors should clarify this interpretation if they intended to include this gene within their gene set.

In the GSEA analysis, the authors set the cut-off of FDR as less than 0.25. However, that value was not statistically appropriate as the cut-off of FDR. Also, this result was so important because this pathway analysis affected the basis of their conclusions.

Additional comments

(Minor problem)
1. Articles referred in this manuscript were so old.
2. The author should write gene names in italics and show the coding protein of those genes.
3. All abbreviations should be first spelled out.

---

## Round 0.3 · Minor Revisions

Thanks for your revised manuscript. The concerns have been addressed in part. But please see my comments below for additional changes that are requested.

1. You have added commentary about the C-index for this model compared to other published articles. Although your C-index is slightly higher than these other studies, you cannot claim that yours is better without formally evaluating that. There are ways to do that, but I assume it is beyond what you want to accomplish here, especially because some of these articles are for other cancer types and you would need to re-analyze those datasets as well. Instead, you should soften the language in this paragraph to indicate that your C-index falls within a similar range as other related studies. Furthermore, as the reviewer suggested, a C-index of 0.65 is likely not good enough to be used in clinical practice. Thus you should remove or modify your claim that "we prove our immune signature to be a promising tool in predicting KIRC patients’ survival outcome." That is a subjective statement. At a minimum, you would need to evaluate this claim on multiple other datasets.

2. I am sorry, but I don't fully understand your response regarding Figure 3. You mention differences between the log-rank test and the univariate Cox analysis, but I am not sure how that addresses the reviewer's concern about lack of significance for GDF1. In addition, you did not address the comment about performing multiple-testing correction.

3. Thank you for justifying the use of the 0.25 FDR threshold.

4. The Tracked Changes version of the manuscript does not show any differences in the References section. Please clarify your statement that you have updated some of the references.

5. Thank you for increasing the font sizes on some of the figures. However, in Figures 3 and 7, there is still text that is far too small. I can see it with a magnifying class only. It's up to you on how to solve this, but one option would be to split these into multiple figures.

6. Thank you for creating Table 4.

7. You do not need to put the data files in GitHub. However, your R script uses different file names than those listed on your site. read.table() can pull the data files directly from Xena Browser. I didn't attempt to run your script yet (because the file names were different), but I will do that when you resubmit to make sure the script will work and that I can reproduce your figures.

---

## Round 0.4 · Minor Revisions

Thank you for the updated manuscript. Below are some additional items for you to address.

* The Methods section states, "The TCGA database was used to collect the clinical and RNA-seq data of 528 KIRC patients." Please be more specific. You should also cite the Xena browser (there is a paper for it).
* Cox regression accounts for covariates, but it doesn't account for multiple tests. The following articles provides some insights on methods of correcting for multiple tests: https://www.nature.com/articles/nbt1209-1135
* The R script uses the following packages, but it appears that some/all of them have not been cited in the manuscript. Please either cite them or remove them from the R script if you did not use them.
circlize, survminer, survivalROC, copynumber, clusterProfiler, maftools, scatterplot3d, survRM2
* I attempted to run the R script but received an error message toward the beginning (see below). I believe the error is because read.table doesn't work with gzipped files. The readr package does have a function to do this. Or there are functions in R to download and gunzip a file. Please make sure the R script runs from beginning to end (and that it repeats your analysis successfully) before resubmitting.

Error in make.names(col.names, unique = TRUE) :
invalid multibyte string 1
In addition: Warning message:
In read.table(file = "https://gdc.xenahubs.net/download/TCGA-KIRC.htseq_fpkm.tsv.gz", :
line 1 appears to contain embedded nulls

---

## Round 0.5 · Major Revisions

Your appeal was forwarded to me regarding the "Reject" decision that I made earlier. I'm willing to reconsider and change this to a "Major Revisions" decision.

First, I will reply to your comments in the appeal. The journal asks that academic editors "comment on the statistics and provide methodological input on whether the statistical aspect is performed to the technical standard required for publication." It is my responsibility to ensure that the quantitative approach is methodologically sound, irrespective of what the reviewers say (or whether I send the article to reviewers at any given time). I understand that there are multiple ways to tackle a given problem. But it is my responsibility to ensure that the methods are rigorous and the results are appropriate to be published. Even though papers that have used similar methods may have been published in other journals previously, that doesn't necessarily indicate that the methods in this paper are ready for publishing.

Please address the following comments. These include some that I made in my previous response, in addition to some new comments.

* If I understand correctly, you evaluated 1,534 genes in the training set for prognostic relevance. When evaluating such a large number of genes, it is likely that you will identify a certain percentage of significant genes by random chance, so you must correct for this large number of tests if you report P-values for these genes on the training set. Multiple testing correction also should be used wherever else you report on multiple related statistical tests. Commonly used methods are the Bonferroni family-wide method or the Benjamini-Hochberg False Discovery Rate method.
* The authors report C-index values, Kaplan-Meier curves, and PCA plots for for the training set and for the combined training+test set. However, these results are subject to overfitting (as with any training set). What matters is the performance on the test set. Please remove all figures that report these results for the training set or training+test sets.
* Please revise the code in your GitHub repository so that other scientists can repeat the analysis. When I last attempted to run the code, it was using some libraries that were not being loaded (or installed) in the code. You mentioned that using Docker might be a way to solve this. Indeed, this can be a great tool for creating an isolated testing environment. I would welcome that. Alternatively, you could test this more simply using a new R session in which no libraries have been loaded and that has an empty environment.
* The Introduction states, "Our immune clusters were more robust and independent." The Introduction is not the best place to make this claim because you have not yet described your methods or results in detail. Also, please clarify in the manuscript what evidence you have to support this claim. Did you perform a side-by-side comparative analysis? Furthermore, you cite a paper by Khadirnaikar, et al. But this paper is not listed in the references.
* The Introduction states about Smith, et al. that "they did not investigate the prognostic ability of the signature in different subtypes of patients." Please clarify what you mean by "different subtypes" and why this is important. Furthermore, the Smith, et al. paper is not in the references.
* The "Establishment of immune-related gene signatures" section needs some clarification. First it mentions a "univariate Cox analysis." It mentions that 650 genes were identified. What criteria were used to conclude that each of these genes had a "prognostic ability"? Is it the P < 0.05 threshold? If so, please state this earlier. Later it mentions that these genes "were evaluated using the Cox proportional hazards model" and that "the gene model with the highest frequency in 1,000 iterations was chosen as the immune-related risk signature." The final model had 14 genes. Please be more detailed in describing how you got from 650 genes to 14 genes. Later it says, "Linear weighing was performed on gene expression value and the Cox coefficient to determine the risk score." Please explain how this works in more detail.
* In the "Assessment of prognostic ability" section, it states, "The independent prognostic ability of the immune-related risk signature was assessed using survival analysis as well as Cox analysis." Cox analysis is a type of survival analysis. Please clarify this.
* Which data did you use for the "Gene set enrichment analysis"? Did you use all samples? Or just the training set? Did you use all genes? Or the 1,534 subset? If you just used the 1,534 subset, how did you account for this in the GSEA analysis?
* The "Statistical analysis" section states, "The student’s t-test was performed for statistical comparison." Where was this used?
* On lines 117-9, it states, "Because of the
differences between the log-rank test and the univariate cox analysis, the results of the univariate cox analysis of other
genes except GDF1 are very consistent with the results of the K-M (Kaplan-Meier) survival curves." I am not sure that I understand exactly what this means. Please clarify in the manuscript.
* On lines 121-123, it says that some genes had high expression, while others had low expression. These are high or low compared to what?
* On lines 128-134, provide more details about the prognostic performance. Although the figures provide numbers that represent the ability to estimate prognoses, it would be helpful to state (at least some of) these numbers in the text. Also, the Discussion provides some information about prognostic ability like, "We found that patients in the high-risk group showed a positive association with M stage, T stage, and advanced pathological staging." This type of information would be relevant in Results.
* On line 169, it states, "demonstrating the clinical significance of our signature and its possible use as a survival predictor in KIRC." The C-index value on the test set was 0.65. Please provide evidence to back up the statement that it has clinical significance. Citing other papers that performed at similar levels is not enough (unless they are already being used in a clinical setting).
* How does the authors' approach compare to what is currently being used by clinicians?
* Figure 7. Please clarify how patients were assigned to the high risk and low risk groups.
* Thank you for increasing the font size in the manuscript figures. The numbers above the green bars in Figure 1 are still extremely small. The P-values in Figure 6 are still extremely small. Much of the text in Figures 4A-B is extremely small.

I will reconsider this article if each of these comments has been addressed carefully.

· Appeal

Appeal


· · Academic Editor

Reject

Dear authors,

I am very sorry, but I have decided to reject this article.

In previous comments, I have asked that you address issues related to multiple testing, but these have not been addressed. If I understand correctly, you evaluated 1,534 genes in the training set for prognostic relevance. When evaluating such a large number of genes, it is likely that you will identify a certain percentage of significant genes by random chance, so you must correct for these tests when reporting P-values. Commonly used methods are the Bonferroni family-wide method or the Benjamini-Hochberg False Discovery Rate method.

In addition to this issue, as I evaluated the article more closely for statistical validity, I became concerned about the model validation approach. The authors report CI index values and Kaplan-Meier curves for for the training set and for the training+test set combined. However, these are invalid because these results are subject to overfitting (as with any training set). The Methods section also lacks details that would provide assurance that the training and test sets were kept completely separate during all steps of preprocessing and external validation.

I attempted to run the updated code but still got an error. It appears the code is using some libraries that are not being loaded (or installed) in the code. The only way to test this properly is to use a new R session in which no libraries have been loaded and that starts with an empty environment. It's not a question of whether someone on the authors' team can run the code. It's a question of whether someone reading this manuscript is able to run the code.

Lastly, I have asked in previous comments that the authors increase the font size for text in the figures. They have done this in some figures, but there are others where the text is so small that it is illegible (e.g., Figures 1 and 4B).

I wish I could give you better news, but I have decided that I cannot move forward with this manuscript.

---

## Round 0.6 · Minor Revisions

I was asked to step in for your appeal and make a recommendation for your submission. Your submission has undergone four rounds of reviews and it appears that you have tried hard to address the comments. I perfectly understand your frustration when a rejection decision was made after such a long review process. While I generally concur with the AE’s comments on the methodological issues, I do believe he should have raised these points at a much earlier stage. Considering all the previous comments and responses, I recommend the following essential revisions before the potential acceptance of your manuscript:

(1) To avoid overfitting, please evaluate your trained model (14 immune-related gene signature) on the test data set only. Please remove all the evaluation results (figures, tables, etc.) based on the training or full (training+test) data set.

(2) In the discussion, please acknowledge the limitation of your evaluation scheme (i.e., randomly dividing the full data set into training and testing data set). Such evaluation is subject to the inherent bias for the specific study (e.g., confounding, batch effects). Ideally, an independent data set should be used for evaluation.

Best,
Jun

---

## Round 0.7 · accepted · Accept

I am happy to accept your revised manuscript, following your appeal.

In the production phase, please modify the following
"‘We acknowledge the limitation of our evaluation scheme, including randomly dividing the full data set into training and testing data set, which result in the inherent bias for the specific study. Another limitation of our research is the lack of independent validation data sets in our evaluation scheme. Therefore, our study should be further validated through a prospective cohort data to further illustrate the robustness of the model’ to
"We acknowledge the limitation of our evaluation scheme, which involves randomly dividing the full data set into training and test data set. Such evaluation is still subject to the inherent bias for the specific study. Therefore, further evaluation through an independent data set is needed to validate our prognostic model."